# ChatGPT- versus human-generated answers to frequently asked questions about diabetes: A Turing test-inspired survey among employees of a Danish diabetes center

**Adam Hulman**[1,2]*, **Ole Lindgård Dollerup**[1], **Jesper Friis Mortensen**[1,2], **Matthew E. Fenech**[3], **Kasper Norman**[1], **Henrik Støvring**[1,4], **Troels Krarup Hansen**[1,5]

1 Steno Diabetes Center Aarhus, Aarhus University Hospital, Aarhus, Denmark, 2 Department of Public Health, Aarhus University, Aarhus, Denmark, 3 Una Health GmbH, Hamburg, Germany, 4 Department of Public Health, University of Southern Denmark, Odense, Denmark, 5 Department of Clinical Medicine, Aarhus University, Aarhus, Denmark

* adahul@rm.dk

**Data Availability Statement:** The data are not publicly available due to institutional data use

## Abstract

Large language models have received enormous attention recently with some studies demonstrating their potential clinical value, despite not being trained specifically for this domain. We aimed to investigate whether ChatGPT, a language model optimized for dialogue, can answer frequently asked questions about diabetes. We conducted a closed e-survey among employees of a large Danish diabetes center. The study design was inspired by the Turing test and non-inferiority trials. Our survey included ten questions with two answers each. One of these was written by a human expert, while the other was generated by ChatGPT. Participants had the task to identify the ChatGPT-generated answer. Data was analyzed at the question-level using logistic regression with robust variance estimation with clustering at participant level. In secondary analyses, we investigated the effect of participant characteristics on the outcome. A 55% non-inferiority margin was pre-defined based on precision simulations and had been published as part of the study protocol before data collection began. Among 311 invited individuals, 183 participated in the survey (59% response rate). 64% had heard of ChatGPT before, and 19% had tried it. Overall, participants could identify ChatGPT-generated answers 59.5% (95% CI: 57.0, 62.0) of the time, which was outside of the non-inferiority zone. Among participant characteristics, previous ChatGPT use had the strongest association with the outcome (odds ratio: 1.52 (1.16, 2.00), p = 0.003). Previous users answered 67.4% (61.7, 72.7) of the questions correctly, versus non-users' 57.6% (54.9, 60.3). Participants could distinguish between ChatGPT-generated and human-written answers somewhat better than flipping a fair coin, which was against our initial hypothesis. Rigorously planned studies are needed to elucidate the risks and benefits of integrating such technologies in routine clinical practice.

policy and concerns about participants' privacy. Steno Diabetes Center Aarhus is part of the Central Denmark Region, whose Legal Department registered our project (survey-based studies do not require ethical approval) and issued an agreement including the data handling and sharing rules. To be able to share the data with any third party, they have to get an approval by the Danish Data Protection Agency, which would not be feasible with a publicly available dataset. Please contact the first author for more information. IRB/Data ethics committee contact info: Legal Department of Central Denmark Region Skottenborg 26 Viborg 8800, DK e-mail: kontakt@rm.dk tel.: +45 78410000

**Funding:** AH, OLD, JFM, KN, HS, and TKH are employed at Steno Diabetes Center Aarhus that is partly funded by a donation from the Novo Nordisk Foundation (no. NNF17SA0031230). AH is supported by a Data Science Emerging Investigator grant (no. NNF22OC0076725) by the Novo Nordisk Foundation. The funders had no role in the design of the study.

**Competing interests:** The authors have declared that no competing interests exist.

## Introduction

The capabilities of artificial intelligence (AI) became more apparent to the general public after OpenAI (San Francisco, CA, USA) released ChatGPT on Nov 30, 2022. ChatGPT is a large language model (also known as a chatbot), that was optimized for dialogue [1]. ChatGPT had 100 million active users in January, 2023, making it the fastest-growing consumer application ever [2]. Potential applications of ChatGPT have also received attention from the medical research community with numerous editorials and commentaries published on the topic in major scientific journals, while some authors already used and attributed co-authorship to ChatGPT [3–6]. Recent studies have demonstrated that ChatGPT and similar AI-based systems reached a level to be able to pass components of the United States Medical Licensing Exam [7–9] and can answer questions in specific medical specialties like genetics and ophthalmology [10, 11]. Such proof of clinical knowledge is promising, but studies and applications in patient-centered scenarios are still rare.

Individuals suffering from chronic conditions such as diabetes dedicate a significant amount of effort and resources towards managing their disease and seeking information, including from online sources. The use of chatbots is not unprecedented. In Norway, the first chatbot integrated into the national health service platform was developed to inform and empower women with gestational diabetes [12]. Our previous work suggests that the majority of patients are open to AI-based solutions, with higher acceptance rates in low-risk situations [13]. This calls for studies investigating the knowledge and value of large language models, such as ChatGPT, in everyday, practical scenarios and routine clinical care. To provide a solid basis in such a process, a rigorous study design and the involvement of healthcare professionals is crucial.

Inspired by the Turing test and non-inferiority trials, the aim of our study was therefore to investigate ChatGPT's knowledge in the diabetes domain, specifically its ability to answer questions frequently asked by patients in a way that is indistinguishable from human expert answers.

We hypothesized that participants (employees of a large Danish diabetes center), who have anything between some and expert knowledge about diabetes, will not be able to distinguish between answers written by humans and generated by ChatGPT in response to frequently asked questions about diabetes. Our secondary hypothesis is that people with contact with patients as healthcare professionals, and those who previously tried ChatGPT, might be better at identifying answers generated by AI.

## Methods

A detailed study protocol was developed following the CHERRIES checklist for e-surveys [14], and published online when the data collection started [15]. Alterations from the protocol were reported previously [15].

### Study population

All employees (full- or part-time) of Steno Diabetes Center Aarhus (SDCA) represented the study population. SDCA is a specialized diabetes center located in Aarhus, Denmark, integrating a clinic with diabetes research facilities and education.

### Survey structure

The first part of the survey included questions on the following participant characteristics: age (<30, 30–39, 40–49, >50), sex (male/female), whether they ever had contact with people with diabetes as a caregiver (yes/no), ever heard of ChatGPT (yes/no), ever used ChatGPT (yes/no; only asked if participant had heard of ChatGPT).

The second part included ten multiple-choice questions with two answers each. One of them was an answer to the question written by a human-expert, while the other was generated by ChatGPT. The order of questions was randomized once, but was the same for all individuals, whereas the order of answers was randomized for each individual and question. Participants were asked to identify the answer they most believe was generated by AI.

The language of the survey was Danish. An English translation was published with the study protocol [15].

## Survey questions

The ten questions were defined to cover five common topics with two questions for each: pathophysiology, complications, treatment, diet, physical activity. Eight questions were identified among the 'Frequently Asked Questions' on the website of the Danish Diabetes Association (diabetes.dk, accessed on Jan 10, 2023), an interest organization for people living with diabetes, and the largest patient association in Denmark. The remaining two questions were formulated by the authors so that they correspond to specific paragraphs from the 'Knowledge Center for Diabetes' website (videncenterfordiabetes.dk, accessed on Jan 10, 2023), and a consensus report on exercise in type 1 diabetes [16].

## Human expert-written answers

The answers were taken directly from the source websites/materials where the questions were identified. For feasibility reasons, some of the answers were shortened by two authors, both healthcare professionals, to fit our target length (45 to 65 words).

## AI-generated answers

After finalizing the questions and the human answers, we used ChatGPT (version Jan 9, 2023) to generate answers by AI. Before including the question, the context and three examples (randomly selected from 13 question-answer pairs) were given to ChatGPT in the same prompt. Few-Shot prompting, i.e. giving input-output pairs to demonstrate examples and their expected format has been shown to be successful to tailor large language models to a given context [17]. The exact prompt used was published in the study protocol. Each question was asked in an independent chat window to avoid information leakage between questions.

Although ChatGPT was instructed to give answers between 45 and 65 words, similar to the human answers, the average number of words was 70 (range: 55–85). The last sentence was often a recommendation to consult a healthcare professional, which we removed in seven cases, where the corresponding human answer did not include such information. We also removed the first sentence in three cases, where it was essentially a repetition of the question. Moreover, we corrected four grammatical mistakes. With this approach, we ended up with 56 words on average, which was four more than for the human answers. The study protocol was updated to highlight these edits [15]. Two answers included incorrect information, however, in the context of the rest of the answer, and the fact that the survey was not conducted among patients, we decided to keep these in the survey. Study participants were informed about these after the data collection had been completed.

## Survey administration

The study was conducted as a closed, web-based survey. Participants were invited by e-mail including person-specific links that allowed them to fill out the survey once. Information about the study was disclosed on the opening page. Participation was voluntary without any

incentives offered. Data was collected in a 96-hour period from January 23 (12:00 CET) to January 27 (12:00 CET), 2023.

### Ethics statement

The study was registered in the database of research projects in the Central Denmark Region (no. 1-16-02-35-23). Participants were informed on the opening page of the survey that they are participating in a research study and by submitting their answers, they give digital consent to contributing with data to the study. Further ethical approval was not necessary in Denmark as the study only included survey-based data collection. The survey was developed and distributed in SurveyXact (Rambøll, Copenhagen, Denmark) that complies with GDPR regulations.

### Statistical analysis

The study was designed similar to non-inferiority trials with the aim of demonstrating that participants could not identify the AI-generated answers better than just flipping a fair coin, which is equivalent to a probability of 50%. Based on a priori precision calculations using simulations, we defined a 55% probability as inferiority margin providing approximately 85% power in case of 100 participants.

A binary outcome was defined as identifying the correct answer (1 if the ChatGPT-generated answer was identified, 0 otherwise). Data was analyzed at question-level using logistic regression models with robust variance estimation with clustering at participant level to account for within participant correlation of answers. Estimated model coefficients of log-odds with 95% confidence intervals (CIs) were transformed into probabilities of correctly identifying AI-generated answers. To test the impact of participant characteristics on these estimated probabilities, first, we fitted univariable models including one characteristic at a time. Finally, we fitted a multivariable model adjusting for all four characteristics i.e. age, sex, patient contact, ChatGPT use, at the same time. This last analysis was planned after the publication of the study protocol, due to the imbalance of participant characteristics. Our power calculations suggested that we would be able to show a difference of at least 9% in probabilities (59% vs 50%) between balanced groups (1:1) and 15% between imbalance groups (9:1) with 90% power and 0.05 alpha [15]. An exploratory analysis was planned using random-effects logistic regression to assess between-person variation. Based on this, we report a 95% prediction interval for probabilities of individuals in our study to answer correctly as a measure of between person variation (normal-based calculation on log-odds scale and then transformed with inverse logit).

The statistical analyses were conducted in R software (R Foundation for Statistical Computing, Vienna, Austria; version 4.2.2) using the *miceadds* (version 3.16–18), *lme4* (version 1.1–31), *Epi* (version 2.47) packages.

## Results

Fifteen invited individuals were registered with two different e-mail addresses, and therefore we had a total number of 311 unique potential participants. Altogether, 183 participants completed the survey (59% response rate).

There were 129 women (70%) and 52 men (28%) out of 183 participants (Table 1). Two participants did not disclose their sex, therefore they were excluded from the sex-specific descriptive statistics and the analyses where this information was necessary. More than half of the participants (107 out of 183, 58%) have had contact with diabetes patients as healthcare providers. Regarding familiarity with ChatGPT, 117 out of 183 (64%) reported having heard of

**Table 1. Study participant characteristics by sex and overall.**

| | Men (n = 52)[a] | Women (n = 129)[a] | Overall (n = 183) |
|---|---|---|---|
| **Age** | | | |
| under 30 years | 5 (10%) | 11 (9%) | 18 (10%) |
| 30–39 years | 23 (44%) | 47 (36%) | 70 (38%) |
| 40–49 years | 12 (23%) | 44 (34%) | 56 (31%) |
| over 50 years | 12 (23%) | 27 (21%) | 39 (21%) |
| **Patient contact as HCP**[b] | 37 (71%) | 68 (53%) | 107 (58%) |
| **Heard of ChatGPT** | 45 (87%) | 72 (56%) | 117 (64%) |
| **Used ChatGPT** | 25 (48%) | 10 (8%) | 35 (19%) |

[a]two participants have not disclosed their sex
[b]HCP: health care professional

it, and 35 of them had used it (19% of all participants). These numbers were even higher among men.

The estimated probability of identifying the AI-generated answer was 59.5% (95% CI: 57.0, 62.0), which was outside of the pre-defined non-inferiority zone (<55%) (Fig 1). We did not find evidence for an association between age and the outcome. Men had a 63.5% (58.6, 68.2) probability of identifying the AI-generated answer, which was higher than for women (57.8% (54.9, 60.6) and p = 0.047 for sex difference). Participants with contact to patients had a probability of 61.2% (57.8, 64.5) of correctly answering the questions compared to 57.3% (53.6, 60.8) for those without patient contact (p-value for difference = 0.12). Those who had used ChatGPT before were more likely to correctly identify the AI answer compared to those who had not (67.4% (61.7, 72.7) versus 57.6% (54.9, 60.3), odds ratio (OR) 1.52 (1.16, 2.00), p = 0.003). An odds ratio of a similar magnitude was found in the combined model, where age >50 years vs. 30–39 years was also associated with a higher probability of correctly identifying the AI-generated answer (OR: 1.30, (1.01, 1.66)).

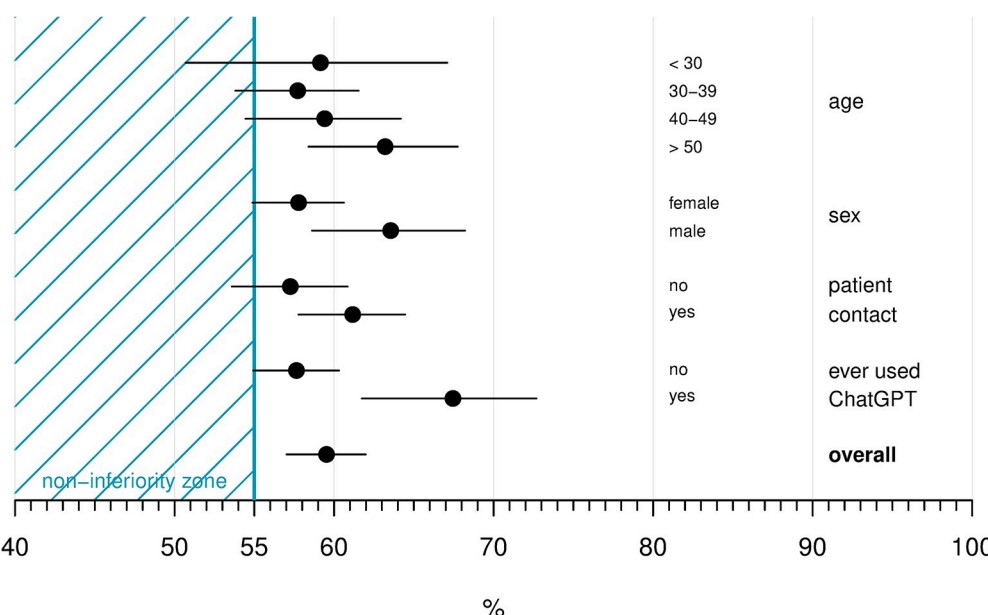

**Fig 1. Probabilities with 95% confidence intervals of correctly identifying the AI-generated answers overall and by participant characteristics.** Presented probabilities were estimated from univariable models.

The proportion of correct answers ranged between 38% and 74% across the ten questions (see S1 File) Estimates and their 95% CIs were in the non-inferiority zone for two questions, overlapped with the inferiority margin (inconclusive result) for another two, and were in the inferiority zone for the remaining six. We also observed a significant variation between individuals identifying the correct answers (95% prediction interval: 46.2%, 71.9%).

## Discussion

Inspired by the Turing test and non-inferiority trials, we conducted an e-survey among all employees of a Danish diabetes center to investigate ChatGPT's ability to answer frequently asked questions about diabetes. Participants could distinguish between ChatGPT- and human-generated answers somewhat better than flipping a fair coin. We found that individuals who had previously used ChatGPT could more often distinguish ChatGPT-generated answers from human answers, while we did not find such strong evidence for a difference by having contact with patients versus not. This may suggest a stronger predictive value of linguistic features rather than the actual content.

Familiarity with ChatGPT was common among participants. Participants who had even used it before could on average reveal 10% more answers correctly than those who had not, suggesting that the structure of the text provided an important clue. Moreover, ChatGPT used some words (e.g. 'libido') that are rarely used in the Danish language, especially in the medical context. This question was the one that participants were most likely to correctly identify the ChatGPT-generated answer (74%). We consider it likely that it would have been even more difficult to distinguish between ChatGPT-generated and human-written answers if the survey had been conducted in English with native English-speaking participants. Also, our study participants were asked to reveal AI-generated answers, which provided extra motivation for them, which would not be the case in everyday use.

We identified only one study with a similar design to ours [18]. The authors extracted patient-provider communication from electronic health records and presented five cases with provider-written answers and another five with ChatGPT-generated answers. Participants, who were recruited online, could identify 65% of AI-generated answers correctly, which is of similar magnitude to our findings, although the authors neither have formally tested any hypotheses, nor presented precision estimates for the results. The survey also asked participants whether they would trust chatbots in different healthcare scenarios, and found an inverse association between trust and the medical complexity of cases.

More studies and the scientific discourse have mostly focused on ChatGPT's role in medical research (e.g. scientific writing [6, 19, 20]), formal testing of large language models' medical knowledge e.g. by taking medical licensing exams [7–9], or answering questions in highly specialized topics [10, 11]. A potential role of ChatGPT in chronic disease management, like diabetes, is yet to be debated. The only evidence we identified was a short report by Sarraju et al., where ChatGPT gave 21 appropriate answers to 25 questions about cardiovascular disease prevention [21].

Although the internet seems to be the preferred source of information about diabetes, patients still seek confirmation from their healthcare professionals [13, 22]. Future efforts need to consider how AI-based solutions can supplement routine care. A good example for this is Dina, a chatbot developed to inform and empower women with gestational diabetes. Based on a 20-week data collection, the authors reported that the chatbot could answer 89% of almost 3,000 questions, providing women with readily accessible information about their condition [12]. Integration of such applications in national digital health platforms governed by the public sector might increase public trust needed for successful implementation. Considering the

dominance of the private sector in the field of AI, and that people are more willing to share their data for AI research with the public sector than with private companies [13], it is relevant to make partnerships between different actors and stakeholders, and patient organizations.

Large language models are powerful tools with a potential to deliver information to millions or even billions of people as next-generation search engines. In addition to the positive impact of innovative digital solutions, this poses also a huge risk of spreading misinformation, which has to be considered critically and handled responsibly. In our study, we observed that two ChatGPT-generated answers included incorrect information. One of them described gestational diabetes as a form of type 2 diabetes. Although both are characterized by reduced insulin sensitivity, gestational diabetes is considered as a separate condition. This type of misinformation would not put patients at serious risk, but if recognised as misinformation by the patient, would have a negative impact on their trust in subsequent answers provided by ChatGPT. In another answer, ChatGPT described the association between intensity of physical activity and blood glucose levels in type 1 diabetes in the opposite direction to what scientific evidence shows, i.e. low-intensity training leading to reduced blood glucose, while high-intensity training to increased [16]. This type of misinformation can have serious consequences to the patients' wellbeing. However, ChatGPT's answer also included the importance of monitoring blood glucose levels before, during and after training, which reduces the risk of potential harm. Nevertheless, this example demonstrates the importance of built-in safeguards in AI-based applications. The role of chatbots in the management of diabetes and other chronic conditions, given their inherent complexity and the constant requirement for highly-personalised decision-making on the part of patients and their carers, remains unclear and deserves deeper scrutiny.

In keeping with this, increasing attention is being paid to the need for, and optimal way to, regulate the use of large language models, both in the healthcare context and more widely. The general risk of misinformation generated by large language models is all the more critical when healthcare-relevant information is being considered. It has been pointed out that much of the biomedical literature would not be available as training data for ChatGPT and similar models, as it exists behind paywalls, and that the rapid progress of science would require frequent retraining of these models [23]. ChatGPT also has a well-documented ability to 'hallucinate', i.e. to make up completely false information supported by fictitious citations. In spite of these issues, there is a regulatory gap around the use of these models, or indeed other forms of generative AI. A legal framework for AI regulation proposed by the European Commission, whose objective is to "guarantee the safety and fundamental rights of people and businesses when it comes to AI" [24], has been touted by EU officials as addressing concerns around large language model-generated misinformation, but has been criticized for putting the onus on AI developers to undertake risk assessment [25], and for conflating risk acceptability with trustworthiness [26]. Moreover, although the use of software tools for medical purposes is usually the domain of medical device regulation, these regulations are usually not considered to be applicable to tools used to provide generic medical information to patients or the general public outside of the strict clinical setting. Thus, guidance on the safe and ethical use of such tools is to date largely confined to non-binding guidelines, frameworks and standards [27]. These all focus on the application of a largely consistent set of 'ethical principles', such as the prioritization of algorithmic fairness and the minimization of bias, to the development of AI tools in healthcare, but many have struggled to move beyond a conceptual discussion of aspiration principles, towards a pragmatic, measureable, and therefore enforceable set of recommendations for the use of this technology [28, 29].

One feature that could strengthen trust would be if large language models provided trustworthy references supporting their answers. Atlas, a large language model developed for

question answering and fact checking, attempts to address this issue by retrieving different sources and synthesizing their content [30]. Such AI models could also be more suitable for adaptation to a specific domain, e.g. diabetes management, and the local healthcare sector and national guidelines.

Although ChatGPT is optimized for dialogue, this happens in a rather one-sided manner so far. A healthcare professional would often reply with questions to collect more information from the patient before making an informed decision or giving an answer. Instead of asking clarifying questions in case of ambiguous questions, ChatGPT guesses what the user meant, a limitation acknowledged by the developers. This important aspect also has to be considered when developing large language models to mimic patient-healthcare professional interactions.

We are among the first to investigate the capabilities of ChatGPT in patient-centered guidance of chronic disease management. A major strength of our study is the combination of expeditious, but scientifically rigorous planning and execution in a clinical setting, allowing us to test pre-defined hypotheses and compare results by participant characteristics. Our study focused on a topic with a potential impact on patient care and not only on the theoretical capabilities of AI. In our investigation, we involved all employees of a diabetes center, with more than half of them acting as care providers.

Our study also has limitations. Due to feasibility and time considerations, we did not conduct a study among patients, who are a key target audience for future clinical applications of large language models with dialogue capabilities. We included only ten questions in the survey, therefore some relevant topics could not be covered. This was driven by a need to find the right balance between the time needed to fill out the survey and the potential to gain insights. Our decision was supported by precision calculation using simulations. The correctness of the answers was only assessed by the authors and was not part of the survey. Although, most questions and corresponding human-written answers were identified on a website with people with diabetes as their main target audience, we cannot exclude the possibility that some participants were familiar with the sources and recognized the human-written answers. Moreover, an average user might not be able to extract the same level of knowledge from ChatGPT, as we did in an experimental setting using few-shot prompting. Other state-of-the-art language models specifically tuned for the medical domain exist (e.g. Med-PaLM [9]), but they are either not openly available, or are not as accessible to the general public as ChatGPT. Also, the use of benchmark datasets and a recently introduced framework for human evaluation will contribute to building a solid foundation for the field [9].

Large language models have shown impressive results in the last few years in various domains of medicine, but their significance was mostly discussed in the data science community while healthcare professionals mostly considered them as mysterious black boxes. ChatGPT's relevance lies in enabling millions of people to interact with state-of-the-art AI. In our study, around 20% of healthcare professionals had tried ChatGPT and almost 70% had heard of it. Applications enabling broader communities to interact with AI might contribute to a better understanding of the potential in AI for healthcare and ultimately to building bridges between data science and medical research. This process, combined with patient and stakeholder involvement is needed for the development of innovative AI-based solutions driven by actual clinical needs.

In conclusion, given the widespread attention being paid to large language models in a variety of applications, it is inevitable that patients and healthcare professionals question how they may contribute to disease management and support practical aspects relevant to patients' everyday lives, even though they are not specifically trained for medical tasks. Large language models optimized for healthcare use are warranted, but rigorously planned studies are needed to elucidate the risks and benefits of integrating such technologies in patient care.

## Supporting information

**S1 File. RMarkdown document.** The R code for processing data and generating the results is compiled as an R Markdown document.
(PDF)

## Acknowledgments

The authors are grateful to all employees of Steno Diabetes Center Aarhus who showed interest and participated in our study.

## Author Contributions

**Conceptualization:** Adam Hulman, Ole Lindgård Dollerup, Jesper Friis Mortensen, Henrik Støvring, Troels Krarup Hansen.

**Data curation:** Adam Hulman.

**Formal analysis:** Adam Hulman, Henrik Støvring.

**Funding acquisition:** Adam Hulman.

**Investigation:** Adam Hulman, Ole Lindgård Dollerup, Jesper Friis Mortensen, Kasper Norman, Henrik Støvring, Troels Krarup Hansen.

**Methodology:** Adam Hulman, Ole Lindgård Dollerup, Jesper Friis Mortensen, Kasper Norman, Henrik Støvring, Troels Krarup Hansen.

**Project administration:** Adam Hulman.

**Resources:** Adam Hulman, Troels Krarup Hansen.

**Software:** Adam Hulman.

**Supervision:** Adam Hulman.

**Validation:** Adam Hulman, Henrik Støvring.

**Visualization:** Adam Hulman.

**Writing – original draft:** Adam Hulman, Matthew E. Fenech.

**Writing – review & editing:** Adam Hulman, Ole Lindgård Dollerup, Jesper Friis Mortensen, Matthew E. Fenech, Kasper Norman, Henrik Støvring, Troels Krarup Hansen.

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
