## [Decision Letter · Decision Letter 0]

19 Jun 2023

PONE-D-23-12539ChatGPT- versus human-generated answers to frequently asked questions about diabetes: a Turing test-inspired survey among employees of a Danish diabetes centerPLOS ONE

Dear Dr. Hulman,

Thank you for submitting your manuscript to PLOS ONE. After careful consideration, we feel that it has merit but does not fully meet PLOS ONE’s publication criteria as it currently stands. Therefore, we invite you to submit a revised version of the manuscript that addresses the points raised during the review process.

We look forward to receiving your revised manuscript.

Kind regards,

Jafar Kolahi

Academic Editor

PLOS ONE

Additional Editor Comments:

The use of chatbots in healthcare has become increasingly popular in recent years. These automated systems are designed to provide patients with quick and easy access to medical information and support. However, for patients with diabetes, the use of chatbots can be hazardous.

Diabetes is a chronic condition that requires careful management to prevent complications. Patients with diabetes need to monitor their blood sugar levels regularly, make dietary adjustments, and take medication as prescribed by their doctor. Chatbots may not be able to provide accurate and personalized advice on these matters, which can be dangerous for patients.

Furthermore, chatbots may not be equipped to handle emergencies or urgent situations that require immediate medical attention. Patients with diabetes may experience sudden drops in blood sugar levels, which can be life-threatening if not treated promptly. In such cases, relying on a chatbot for help can delay necessary medical intervention.

However, while chatbots have their benefits in healthcare, they may not be suitable for patients with diabetes. Patients with this condition should consult with their healthcare provider for personalized advice and support to manage their condition safely and effectively. Hence, please add a paragraph to the discussion section about the hazards of using AI for diabetic patients.

Reviewers' comments:

Reviewer's Responses to Questions

**Comments to the Author**

1. Is the manuscript technically sound, and do the data support the conclusions?

Reviewer #1: Partly

Reviewer #2: Partly

2. Has the statistical analysis been performed appropriately and rigorously? 

Reviewer #1: Yes

Reviewer #2: Yes

3. Have the authors made all data underlying the findings in their manuscript fully available?

Reviewer #1: Yes

Reviewer #2: No

4. Is the manuscript presented in an intelligible fashion and written in standard English?

Reviewer #1: Yes

Reviewer #2: Yes

5. Review Comments to the Author

Reviewer #1: The authors describe a ChatGPT Turing Test to determine whether participants would be able to discriminate whether answers to questions about diabetes were produced from humans or generated from ChatGPT. They secondarily hypothesized that participants with ChatGPT experience would be better at identifying ChatGPT produced answers. Participants were employees at the Steno Diabetes Center Aarhus.

Q1: As the authors state, human answers were taken directly from source websites/materials. How do you account for possible participant familiarity/memory of these materials, which could identify human responses?

Q2: How do the authors account for different levels of education among this participants with respect to diabetes, and could this affect responses?

Q3: Given that the survey was web based, and some of the materials are web based, how do you account for participants performing an internet search for the questions, and finding the exact or near exact answers online?

Q4: Given the occasional confabulation by ChatGPT, is it possible that experts on diabetes who took the survey were able to spot these errors to identify the ChatGPT answer? The authors identify 2 examples of this in their survey.

Reviewer #2: Thank you for your interesting and rigorous work. The manuscript is well written and the statistical approach is technically sound. The question is relevant to health care practice. Suggestions to improve the manuscript are provided below.

1) The authors do not provide publicly available data due to institutional (Aarhus University Hospital) data use policy and concerns about participants' privacy. It needs to be evaluated if this paper could be considered an exception to PLOS Data policy.

2) Consider adding further characterisation of study participants regarding professional roles (administrative staff? nurses? doctors? other health professionals?).

3) It is stated that the utility of the answers was assessed by the authors and was not part of the survey. This analysis would be relevant to present, specially if conducted (even if post-hoc) by a blinded outcome assessor. It is stated that 2 AI-generated answers included incorrect information, but there is no correctness information regarding human-generated answers. The utility analysis could include correctness, as this would be relevant to the study's aim (to investigate ChatGPT’s knowledge in the diabetes domain).

4) There is no mention to conformity to STROBE guidelines, as required by PLOS One Submission Guidelines for observational studies.

5) The prompts and examples given to ChatGPT could bias the study in favour of non-inferiority. Although the rationale for the prompts is clear, consider stating it as a limitation in the Discussion section.

6) It is stated in the abstract that "Participants could distinguish between ChatGPT-generated and human-written answers somewhat better than flipping a fair coin. However, our results suggest a stronger predictive value of linguistic features rather than the actual content.". Consider making it clearer in the conclusion that the results did not support your initial hypothesis, as stated in the study protocol (we hypothesized that participants who have at least some and up to expert knowledge about diabetes, will not be able to distinguish between answers written by humans and generated by AI in response to diabetes-related questions), and that the non-inferiority margin was reached. I am unsure if the results strongly support the stronger predictive value of linguistic features (considering the AI's incorrect statements), and would suggest keeping this sentence in the Discussion, but retracting it from the abstract.

6. PLOS authors have the option to publish the peer review history of their article (what does this mean?). If published, this will include your full peer review and any attached files.

Reviewer #1: No

Reviewer #2: No

---

## [Author Response · Author response to Decision Letter 0]

24 Jul 2023

Additional Editor Comments:

The use of chatbots in healthcare has become increasingly popular in recent years. These automated systems are designed to provide patients with quick and easy access to medical information and support. However, for patients with diabetes, the use of chatbots can be hazardous.

Diabetes is a chronic condition that requires careful management to prevent complications. Patients with diabetes need to monitor their blood sugar levels regularly, make dietary adjustments, and take medication as prescribed by their doctor. Chatbots may not be able to provide accurate and personalized advice on these matters, which can be dangerous for patients.

Furthermore, chatbots may not be equipped to handle emergencies or urgent situations that require immediate medical attention. Patients with diabetes may experience sudden drops in blood sugar levels, which can be life-threatening if not treated promptly. In such cases, relying on a chatbot for help can delay necessary medical intervention.

However, while chatbots have their benefits in healthcare, they may not be suitable for patients with diabetes. Patients with this condition should consult with their healthcare provider for personalized advice and support to manage their condition safely and effectively. Hence, please add a paragraph to the discussion section about the hazards of using AI for diabetic patients.

Thank you for your comments on our manuscript. We aimed to address safety issues in the manuscript from a broader clinical perspective (paragraph #6-7 in the Discussion), but following the suggestion of the Editor, we added a more diabetes and chronic disease specific sentence to paragraph #6. 

‘This type of misinformation can have serious consequences to the patients’ wellbeing. However, ChatGPT’s answer also included the importance of monitoring blood glucose levels before, during and after training, which reduces the risk of potential harm. Nevertheless, this example demonstrates the importance of built-in safeguards in AI-based applications. The role of chatbots in the management of diabetes and other chronic conditions, given their inherent complexity and the constant requirement for highly-personalised decision-making on the part of patients and their carers, remains unclear and deserves deeper scrutiny.’

We believe that the rigorous planning of our study sets an example for future studies to evaluate the hazards of large language models applied in healthcare.

Reviewer #1: The authors describe a ChatGPT Turing Test to determine whether participants would be able to discriminate whether answers to questions about diabetes were produced from humans or generated from ChatGPT. They secondarily hypothesized that participants with ChatGPT experience would be better at identifying ChatGPT produced answers. Participants were employees at the Steno Diabetes Center Aarhus.

We appreciate the reviewers' comments and questions. Please find our point-by-point answers below.

Q1: As the authors state, human answers were taken directly from source websites/materials. How do you account for possible participant familiarity/memory of these materials, which could identify human responses?

The majority of the questions were from the Danish Diabetes Association’s website, more specifically the Frequently Asked Questions page, as described in the study protocol and the manuscript. 

‘Eight questions were identified among the ‘Frequently Asked Questions’ on the website of the Danish Diabetes Association (diabetes.dk, accessed on Jan 10, 2023), an interest organization for people living with diabetes, and the largest patient association in Denmark.’

The Danish Diabetes Association is a patient organization and as such the main target audience of the FAQ page is patients. Therefore, we don’t believe that familiarity with the materials could have a major impact on the results, but of course we cannot exclude this either.

We added the following sentence to the paragraph on limitations of our study.

‘Although, most questions and corresponding human-written answers were identified on a website with people with diabetes as their main target audience, we cannot exclude the possibility that some participants were familiar with the sources and recognized the human-written answers.’ 

Q2: How do the authors account for different levels of education among this participants with respect to diabetes, and could this affect responses?

We asked participants the simple question whether they have contact with patients as healthcare professionals, which is the closest proxy for different levels of knowledge about diabetes. We compared these groups with regard to the outcome, which is reported in the Results section and on Figure 1. We agree with the reviewer in that this could definitely affect the outcome, however, many participants were researchers with extensive knowledge about diabetes even without patient contact, which could have diluted the differences. It is impossible for us to investigate this further, as we cannot assess retrospectively the participants’ knowledge about diabetes. 

Q3: Given that the survey was web based, and some of the materials are web based, how do you account for participants performing an internet search for the questions, and finding the exact or near exact answers online?

On the opening page of the survey, participants were asked not to use any help (e.g. Google) or too much time on the answers, but rather their intuition. Therefore, we think the risk of this having a major impact on the results is minimal.

Q4: Given the occasional confabulation by ChatGPT, is it possible that experts on diabetes who took the survey were able to spot these errors to identify the ChatGPT answer? The authors identify 2 examples of this in their survey.

The correct answer with regard to gestational diabetes was revealed by 63% of participants (68% among healthcare professionals [HCPs], 57% among others), while 47% chose the correct one about type 1 diabetes and physical activity (56% among HCPs, 35% among HCPs). 

Reviewer #2: Thank you for your interesting and rigorous work. The manuscript is well written and the statistical approach is technically sound. The question is relevant to health care practice. Suggestions to improve the manuscript are provided below.

We thank the reviewer for the positive comment and appreciate the suggestions. Please find our point-by-point comments below.

1) The authors do not provide publicly available data due to institutional (Aarhus University Hospital) data use policy and concerns about participants' privacy. It needs to be evaluated if this paper could be considered an exception to PLOS Data policy.

We have added more information on this issue in our cover letter. According to our current agreement, the Legal Department of the Region of Central Denmark requires approval from the Danish Data Protection Agency for third parties to get access to the data, which makes it infeasible to share the dataset publicly. We have addressed this issue in the cover letter of the revision.

2) Consider adding further characterisation of study participants regarding professional roles (administrative staff? nurses? doctors? other health professionals?).

It is a valid point by the reviewer, but unfortunately, our agreement with the Legal Department does not allow us to link the dataset to any administrative database about the participants.

3) It is stated that the utility of the answers was assessed by the authors and was not part of the survey. This analysis would be relevant to present, specially if conducted (even if post-hoc) by a blinded outcome assessor. It is stated that 2 AI-generated answers included incorrect information, but there is no correctness information regarding human-generated answers. The utility analysis could include correctness, as this would be relevant to the study's aim (to investigate ChatGPT’s knowledge in the diabetes domain).

Thank you for pointing this out. We meant correctness instead of utility, as the latter heavily depends on the context and therefore is difficult to assess, which we now revised in the Methods section. All human-written answers were considered correct.

4) There is no mention to conformity to STROBE guidelines, as required by PLOS One Submission Guidelines for observational studies.

Instead of the STROBE guidelines, we applied the CHERRIES reporting guidelines. This is specifically developed for web-based surveys and is also part of the Equator Network.

5) The prompts and examples given to ChatGPT could bias the study in favour of non-inferiority. Although the rationale for the prompts is clear, consider stating it as a limitation in the Discussion section.

Few-shot learning is a common approach in extracting and assessing knowledge encoded in large language models (Singhal et al. 2022, arxiv.org/abs/2212.13138). The main rationale behind using it was to get answers in a similar format (length, complexity of language) to those of human answers. However, we acknowledge the difference between knowledge encoded in a large language model, and the knowledge an average user can extract from ChatGPT via its user interface. To clarify this issue raised by the reviewer, we added the following sentence to the Discussion:

 ‘Moreover, an average user might not be able to extract the same level of knowledge from ChatGPT, as we did in an experimental setting using few-shot prompting.’

6) It is stated in the abstract that "Participants could distinguish between ChatGPT-generated and human-written answers somewhat better than flipping a fair coin. However, our results suggest a stronger predictive value of linguistic features rather than the actual content.". Consider making it clearer in the conclusion that the results did not support your initial hypothesis, as stated in the study protocol (we hypothesized that participants who have at least some and up to expert knowledge about diabetes, will not be able to distinguish between answers written by humans and generated by AI in response to diabetes-related questions), and that the non-inferiority margin was reached. I am unsure if the results strongly support the stronger predictive value of linguistic features (considering the AI's incorrect statements), and would suggest keeping this sentence in the Discussion, but retracting it from the abstract.

Thank you for this suggestion. We have revised the abstract to make the main results more clear.

‘Overall, participants could identify ChatGPT-generated answers 59.5% (95% CI: 57.0, 62.0) of the time, which was outside of the non-inferiority zone. … Participants could distinguish between ChatGPT-generated and human-written answers somewhat better than flipping a fair coin, which was against our initial hypothesis. However, our results suggest a stronger predictive value of linguistic features rather than the actual content.’

---

## [Editor Report · Decision Letter 1]

16 Aug 2023

ChatGPT- versus human-generated answers to frequently asked questions about diabetes: a Turing test-inspired survey among employees of a Danish diabetes center

PONE-D-23-12539R1

Dear Dr. Hulman,

We’re pleased to inform you that your manuscript has been judged scientifically suitable for publication and will be formally accepted for publication once it meets all outstanding technical requirements.

Kind regards,

Jafar Kolahi

Academic Editor

PLOS ONE

Additional Editor Comments (optional):

.
---

## [Editor Report · Acceptance letter]

23 Aug 2023

PONE-D-23-12539R1 

ChatGPT- versus human-generated answers to frequently asked questions about diabetes: a Turing test-inspired survey among employees of a Danish diabetes center 

Dear Dr. Hulman:

I'm pleased to inform you that your manuscript has been deemed suitable for publication in PLOS ONE. Congratulations! Your manuscript is now with our production department. 

Kind regards, 

on behalf of

Dr. Jafar Kolahi 

Academic Editor

PLOS ONE